# High Performance SDN WLAN Architecture

**DOI:** 10.3390/s19081880

**Published:** 2019-04-19

**Authors:** Kristián Košťál, Rastislav Bencel, Michal Ries, Peter Trúchly, Ivan Kotuliak

**Affiliations:** Faculty of Informatics and Information Technologies, Slovak University of Technology in Bratislava, 842 16 Bratislava, Slovakia; michal.ries@stuba.sk (M.R.); peter.truchly@stuba.sk (P.T.); ivan.kotuliak@stuba.sk (I.K.)

**Keywords:** WiFi, SDN WLAN, WiFi handover, client mobility

## Abstract

Wireless Local Area Network (WLAN) infrastructure is a dominant technology for direct access to the Internet and for cellular mobile data traffic offloading to WLANs. Additionally, the enterprise infrastructure can be used to provide functionality for the Internet of Things and Machine to Machine scenarios. This work is focused on improvements of radio resources control scalability similar to mobile networks via handover between cells. We introduce an improved IEEE 802.11 architecture utilizing Software-Defined Networks (SDNs). The proposed architecture allows communications during device movements without losing a quality of service (QoS). The fast seamless handover with QoS enables efficient usage of radio resources in large networks. Our improvements consist of integrating wireless management to OpenFlow protocol, separating encryption and decryption from an access point. In parallel, this feature as a side effect unloads processing at the Access Points (APs). Finally, the functionality of architecture design and scalability was proven by Colored Petri Nets (CPNs). The second proof of our concept was performed on two scenarios. The first scenario was applied to a delay sensitive use case. The second scenario considers a network congestion in real world conditions. Client’s mobility was integrated into both scenarios. The design was developed to demonstrate SDN WLAN architecture efficiency.

## 1. Introduction

One of the actual challenges of the Wi-Fi industry is to achieve more efficient management of wireless network ecosystems including a wide range of different segments. This requires dramatic simplification of Wi-Fi network management. Furthermore, an actual state of the technology has already introduced more extensive programmability, automation and machine learning capabilities for Wi-Fi infrastructure [1,2,3]. This brings new opportunities for Wi-Fi network ecosystems, e.g., collecting, processing own analytical network data and acting on them. This trend especially targets enterprise WLAN infrastructure segment; because it will naturally lead to a decrease in network management complexity and increase in reliability and security. The key aspect of the efficient management is providing a client mobility with minimal loss in quality of service. 

The original IEEE 802.11 standard was not primarily focused on providing good client mobility but more focused on network connection. A decision about a client’s handover is performed by client stations which can lead to decreased network performance because a client does not know all the information about network. Another drawback is possibility to make association only to one access point. In the process of performing a handover, the station needs to reassociate to a new access point. The process includes discovery, reauthentication and reassociation phases. The reauthentication time was increased by release of standard IEEE 802.11i. The reduction of the handover time was achieved by release of the standards IEEE 802.11r and IEEE 802.11k but the decision on handover performing is still situated at the client side. 

Our vision is to follow recent trends and provide solution to simplify network management based on SDN WLAN infrastructures [4] and good mobility management. The reasons of selecting the SDN architecture for the concept of a personal access point are: representation of the modern approach in networking; distributed networks use non-standard components for required functionality; simplify solving problem of radio resources control and does not introduce additional problems. The SDN allows one to use standard components for this concept. The personal access point does not require any modification in physical and medium access (MAC) layers. We already presented our initial experiments in two conference papers [5,6] which are extended in this paper. We added formal verification model, Colored Petri Nets, which prove correctness and scalability of the radio resources control for large networks. New real-time environment tests were conducted, and we also added into architecture a new Encryption and Decryption component. The design is improved and described more precisely with definition of all messages and experimenter extensions. 

SDN principles are: decoupling control and data planes; introducing a controller for the management plane; introducing virtualizations of network’s components. Furthermore, the SDN- based solution allows us to introduce multiple SDN applications. Nowadays there are several protocols which can be used as control protocol between the control layer and the infrastructure layer in the SDN e.g., OpenFlow, NETCONF, P4 protocol, etc. In the academic area, the OpenFlow protocol represents the standard in research. The authors in [7] claim that application of the OpenFlow protocol in wireless networks is not suitable from the programmer’s point of view. However, a low level of programmer’s abstraction is not an obstacle to the OpenFlow protocol for the architecture design in the wireless SDN architecture because it allows building a high-abstraction framework for programmers’ packet handling. The framework built on the OpenFlow protocol is able to simplify packet handling for programmers. In this work, we are focused on streamlining network resources for this kind of the framework. 

In our architecture, we decided to propose and verify two features. The first one is the embedded encryption component for centralized end-to-end data encryption within our SDN network. None of the actual state of the technology solutions, except One Big AP [8], have mentioned an embedded encryption method. The One Big AP has not described processes of the embedded encryption and what relations exist between embedded encryption, controller and access point. Also, information on how data are transferred between AP and embedded encryption is missing. As a side effect, an embedded encryption component offloads AP processing. The second feature is improving the management of seamless handover via our extension of OpenFlow protocol. We decided to improve the management of seamless handover because of the following reasons: the client’s mobility is a very critical part of communication for many user scenarios; radio resources control provides better scalability; clear design and transparency using standardized SDN interfaces. Furthermore, we integrate all management messages into one OpenFlow control channel. Advantages of using one control channel are: more transparent management of messaging and easier scalability of network management. Scalability in terms of multi-controller approach (management of wireless part of the network) and its application. We do not directly propose procedures; we just provide means to easily achieve this goal. As we have unified management of wireless and wired part of the network into one channel (managed with the same principle) there is no need to create any new difficult solution how to split mentioned management between many controllers. The integration of messages into OpenFlow protocol simplifies usage of the SDN multi-controller architecture. It is not affecting fast seamless handover and throughput of the network. An example of multi-SDN solution is given in [9]. Solutions supporting more control channels (e.g., Odin [10], Chandelle [11]) have usually lower network performance and slower handover processes. Reaching the mentioned goals allows the usage of WLAN enterprise networks in M2M and IoT scenarios. Examples of usage are for instance uploading flight data after landing, communication between cars within a city or another infrastructure, in trains, etc. [12]. 

Scalability of SDN solutions, e.g., [13], allows further extensions towards SDN applications based on big data, machine learning and artificial intelligence in order to achieve optimal network operating configurations. In this direction our further research is focused on the development of a self-managed WLAN enterprise network for multiple scenarios.

The solution proposed in Section 3 introduces a new architecture which adopts advantages from existing solutions to the new one. One of the key novelties of our proposed architecture is a simplified network management of the wireless part of the network that focuses on optimization of used network resources. This goal is achieved by reducing the number of control channels between control and data plane. Additionally, the architecture increases security by moving encryption functionality to the distributed system (wired part of the network) what has also impact on reduction of AP processing. The proposed architecture is verified by Colored Petri Nets (CPNs) which represent a formal verification that is missing in other solutions. CPN show that the architecture also works in a more scalable network. The architecture is also tested and verified in a real environment. 

Our paper is organized as follows: in Section 2, there is a presentation of an overview of state of the technology solutions for the management of WLAN infrastructures introduced in the last decade. We focused on the explanation of all key principles used for the management of WLAN infrastructures. Section 3 describes a new network architecture, its components, the extension of OpenFlow protocol and process flows. Section 4 describes proof-of-concept validation with analytical tool Colored Petri Nets and measurements on our testbed. The performance of WLAN network management was evaluated under two scenarios. The first one tests the client’s mobility for delay sensitive test case and the second one investigates management performance under heavy traffic. Our source code, materials and methods are described in Section 5. Conclusions are presented in Section 6.

## 2. Related Work

We are aware that not all solutions for management of 802.11 infrastructures are described below but we try to cover all principles for management of 802.11 infrastructures. In some cases, we were able to reference published performance evaluations. Especially parameters of the client’s mobility and data throughput are interesting for us. 

A lot of studies that provide good mobility and good performance, focus on programmability abstraction. The abstraction focuses on mobile station mobility or a new algorithm for mobile management. This interesting approach does not target clearly the architecture design, scalability and relations between managing a wireless part and a wired part of a network. The improved architecture design with well-defined interfaces enables improving frameworks to provide network programmability which results in better quality of service. 

Later in this section, we describe the Odin framework more precisely because it represents approach of using the Virtual Access Point (VAP) concept. Our proposed architecture also uses the VAP concept with some additional improvements which are already mentioned before. We used VAP concept previously but without SDN principles in [14]. The paper describes an architecture for traditional networks. 

### 2.1. CAPWAP

The Control And Provisioning of Wireless Access Points (CAPWAP) protocol [15] was designed for managing and securing wireless access points. It is a combination of personal AP’s Local MAC and Split MAC. The CAPWAP protocol was designed only for wireless networks. It is defined in RFC5415 and it was extended in 2009 to RFC5416, binding only for IEEE 802.11 standards [16].

CAPWAP defines two message classes: management and data class. Data class encapsulates wireless data payload and management class covers whole communication between the Wireless Termination Point (WTP) and Access Controller (AC). These message types are encapsulated in UDP datagrams and each message type has its own UDP port. In order to avoid MTU overflow, CAPWAP defines its own UDP datagram fragmentation. The CAPWAP management messages between WTP and AC are sent by Datagram Transport Layer Security (DTLS) tunnel in a secure mode. DTLS protocol can optionally be used for secure transfer of data messages. The CAPWAP also summarizes statistics on the flow between WTP and station (STA). To manage the communication channel between AC and WTP, the CAPWAP defines keep-alive, a link-keeping mechanism. WTP can detect that AC is not available and can trigger a search for another AC in the network. Deployment of CAPWAP protocol was limited due to its complexity, but on the other hand CAPWAP introduced principles of decoupling data and management traffic.

### 2.2. Chandelle

The recent study on the integration of better management over 802.11 networks performed by the Russian Research Institute for Computer Networks [11] introduced a solution for a smooth and fast Wi-Fi handover, Chandelle. Beyond the OpenFlow protocol in SDN, they integrate CAPWAP as well. They built the architecture by placing SDN over WLANs infrastructure. In their proposed architecture, there are physical SDN forwarders connected to an SDN controller. Each forwarder is connected to a physical router working as an AP. The routers are connected to a new element called the Wireless Access Controller (WAC). Chandelle handover is triggered as follows: the station goes into coverage with other AP or station signal with actual AP weakens, AP sends messages through the CAPWAP to the WAC and it notifies the SDN controller that it should modify the flow table on the forwarders to be ready for handover. The published results show a handover duration approximately 850 ms, which nowadays is significantly outperformed by other solutions. The Chandelle project presents a very complex solution joining SDN and CAPWAP approaches for WLAN management.

### 2.3. Fast BSS Transition IEEE 802.11r-2008 and IEEE 802.11k-2008

The first full implementation of 802.11r [17], which is now part of the 802.11-2016 standard [18], was done at the Queen’s University, Canada [19] in 2008. The goal was to deploy a WLAN infrastructure supporting distributed multiple APs access and fast handover according to the 802.11r specifications. The policy of this standard says that the connection to the new AP is created before the loss of access to the last AP. Standard 802.11r allows storage of cipher keys on all APs for authentication to an Authentication, Authorization, and Accounting (AAA) server. Therefore, handover is simplified only to exchange of authentication messages with the target AP. Handover duration is decreased as a consequence. Experimental results show a significant reduction of handover duration. The duration values of handover in [19] are 50 ms which represents more than a 450 ms reduction against standard WLAN distributed networks. This is a noteworthy result, which is suitable for VoIP applications. However, the decision for handover remains at the client side and thus it cannot provide efficient network management techniques from administrator’s point of view.

The 802.11k [20] improves seamless handover as well. A client station is aware in advance which AP is suitable for next handover including all information how to do it. The client station is connected to an AP which provides the station information about the nearest APs and channels. 802.11r and 802.11k standards are most frequently implemented together [21] so they both share the same disadvantages. 

### 2.4. Proprietary Solutions

Actually, there are many proprietary solutions available for WLAN infrastructure with the seamless handover, but they all are private and closed-source. The most dominant are Aruba—Instant, Cisco—Meraki, Fortinet (Meru)—FortiWifi, Accton. Common features of these solutions are: (i) support for IEEE 802.11 as well as Ethernet network cards; (ii) they do not require changes to terminal equipment (data obtained from terminals are on standard 802.11); (iii) they provide API for node mobility, AP virtualization, WLAN, and QoS (both Wi-Fi and Ethernet); (iv) inspired in concept of personal access point. The common drawbacks of these solutions are limited scalability (we need to use only exact hardware provided by the vendor, which is often very expensive) and management options (mentioned solutions use their own cloud-based management platforms so we cannot use any basic management tools not even manage a device directly via Ethernet).

A good showcase of the proprietary solution is Accton [22] which introduced the concept of a personal virtual access point (PVAP). This is located in the control element and therefore the Wi-Fi network is transformed into an SDN-like structure without changing any hardware or adding an OpenFlow device. Because PVAP for each client station is found as a structure in the control element, the entire PVAP moves between the AP as the client station moves across the network. This functionality is similar for all the mentioned solutions. Accton mentions handover times of around 150 ms, but Cisco and Aruba claim handover times of around 15 ms. Other solutions do not have any information about handover duration times. All of the solutions are closed and private, we do not have any of them for testing purposes, so we cannot check and prove or refute these numbers. Some proprietary solutions have implemented 802.11r and 802.11k standards.

### 2.5. One Big AP

The One Big AP [8] is an architecture in which the authors proposed an illusion of a single AP for the whole WLAN network infrastructure. All APs are transparent and set to the same channel, SSID and BSSID, and introduce to the client station one “big” AP. 

The handover mechanism between APs requires the only exchange of flow tables, which significantly reduces handover duration to an execution of the exchange. The client is unaware of this seamless AP handover. The One Big AP architecture supports two types of devices, an AP called Wireless Access Switch (WAS) and Wireless Backbone Switch (WBS) in the backbone. WBS is OpenFlow-enabled. This work solves mainly seamless handover issues and SDN-based control of the wireless network, and it is the proof of SDN principles for management of WLAN infrastructure. The drawbacks are huge scalability issues and interferences. In the proposed architecture, they proactively modify flow messages inside the topology, and while scaling to large WLANs, there will be a need for more WASs. This action will lead to an increased number of sent flow modify messages by each new WAS in the topology. The One Big AP solution ignores the interferences between APs using a single wireless channel which will make a lot of collisions in large scale networks and thus reduces throughput. 

### 2.6. Odin

Odin [23] is an SDN framework for management of 802.11 networks infrastructures. It integrates enterprise WLAN services: network management, authentication, authorization and accounting, mobility, load balancing, cyber security and many others. In order to simplify client management, Odin introduced a Light Virtual AP (LVAP) which represents a form of the VAP concept. Using LVAP, Odin provides the programmer with a virtual, constant link between the station and AP [24]. Elements of Odin’s architecture are described as follows:
*Light Virtual AP*—It represents the abstract link between the station and the AP. Each physical AP has all the LVAPs that are attached to it. By moving the LVAP from one AP to the other AP, an effective handover is achieved. In essence, each station, thanks to LVAP, thinks it is alone in the network which allows the AP to communicate with the station via unicast. LVAP contains the station’s MAC address, IP address, LVAP SSID, and BSSID that is unique for each station.*Odin Master*—In this case, the OpenFlow application on top of the control element. It is implemented over the Floodlight OpenFlow controller. It can create, add or remove LVAP, request AP statistics, update individual tables, and so on. *Odin Agent*—It is an application over a physical AP. In addition to SDN forwarding tables, it can process LVAP and store information about stations that are connected to it (using radiotap headers). Odin Agents capture probe requests from the stations. In case of capturing probe request message from an unknown station, it sends the message to Odin Master. If LVAP has not been created yet, the Master will create and write it to the Agent. The Agent then responds with a probe response message containing the unique BSSID provided by the Master. Then authentication and association are followed. Authentication is performed by the agent storing the message encryption key into LVAP, which is negotiated with the station. After associating the station, the Agent tells whether the client station has been provided an access to the network.

A handover requires that the Odin Master obtains a station’s statistics from the probe messages or other type messages. These data are compared with statistics from the Agent to which stations are connected. If it detects that a station has better signal strength on other AP, the Master will send a message. Along with an Add message sent to the new AP, a Delete message is sent to the old one.

The Odin handover test was performed with a single controller, two APs, and one station [10]. Data were downloaded during the test session. In their test, there was no disconnection of client stations from the network. In standard topology, disconnecting and reconnecting to the network decreases throughput because the station loses access to the network for some time. In SDN using LVAP, the throughput does not drop. It is due to the fact that each AP only exchanges the station’s LVAP without letting the station know and there is no disconnection from the network, so the data are flowing smoothly.

### 2.7. AeroFlux and OpenSDWN

The next research project defining progress in state of the art is from the German Innovation Laboratories for Telecommunication Networks in 2014 [25]. They created the architecture called AeroFlux which is built upon an Odin framework. The architecture is built on two layers of the control plane. One layer is represented by an element that controls frequent events near the point where they occur, close to the data plane, and that is why it is called Near-Sighted Controller (NSC). A general event that needs a “helicopter” view of the network is controlled by a Global Controller (GC), a logically centralized part of the control layer. As part of the AeroFlux [26] architecture, the Light Virtual Access Points (LVAPs) are defined for each physical AP. LVAPs have stored associations of the station (e.g., each client station has one LVAP) and authentication status, as well as the OpenFlow rules. On the individual APs, a Radio Agent (RA) is installed which processes LVAPs. When the client station handoffs between APs, GC asks NSC to move LVAP to this new AP. The handover process does not need an additional authentication. This is achieved by extended OpenFlow rules called Wireless Datapath Transmission rules (WDTX) which define per-flow 802.11 properties. The test results show transition times of approximately 20 ms, which is a very good result. On the other hand, this approach uses a huge number of devices in the control layer generating a lot of signaling messages and that reduces WLAN data payload throughput. This study is one of the main sources of an inspiration in solving the project and points out that modern times increasingly require WLANs to be centrally managed.

A natural evolution of AeroFlux is OpenSDWN [27]. OpenSDWN uses LVAP from Odin and extends it to NFV and three new features: (i) unified programmability and abstractions of virtualized APs and virtualized middleboxes to handle and migrate per-client state; (ii) programmable datapath to control per-flow wireless transmission settings; (iii) participatory interface to allow users to define priorities and policies. AeroFlux and OpenSDWN define today’s state of the technology for an open source solution for the management of 802.11 architectures.

### 2.8. Summary

The solutions analyzed in this section illustrate the drawbacks and challenges of connecting SDN and IEEE 802.11 technologies which need to be addressed. The scientific challenges are: scalability in means of using any non-vendor specific hardware components and controllers which can be distributed; management options to entirely adjust all controlling to your specific needs and avoid interferences and collisions; fully use OpenFlow standard advantages (e.g., Experimenter messages) to improve security, clean the architecture and make it transparent. SDN security is a big challenge because its programmable aspect presents a complex set of problems to cope with. In this context, wireless networks are more vulnerable than fixed wired networks since broadcast wireless channels easily allow message eavesdropping and injection. Security is minimally specified in SDN; thus, the OpenFlow specification does not describe the security format to ensure data integrity [2]. SDN security will require more sophisticated encryption and authentication mechanisms to prevent hackers. In developing solutions to address the myriad of security challenges in SDNs, we need to cope with one primary challenging issue: managing the trade-offs between the network security and performance. 

In this article we try to target management options, usage of OpenFlow standard specification and security issues. We are not trying to target scalability separately because that challenge is not fully within our scope and we explained why in the Introduction.

## 3. The Architecture of Proposed SDN Network

This section presents the SDN architecture for the IEEE 802.11 wireless infrastructure. It includes architecture design, description of components and explanation of proposed SDN network. Our approach uses a concept of the virtual access point to simplify management and ensure client mobility without a requirement of modification at client side. Our architecture design goals are providing simplified management and client mobility. Moreover, we keep in mind the following goals: maximize 802.11 throughput;efficiently perform SDN management signaling and creating resources to better manage a wireless and a wired part of a network;provide fast seamless handover even for delay sensitive applications;improve WLAN security.

### 3.1. High-level Architecture

Our approach supports three standard layers of the SDN architecture: Application, Control, and Infrastructure. Our contribution is focused on the client’s mobility and authentication that both run at the application layer. Communication between the application layer and the control layer is provided by API. The API modules are running in the SDN controller. Floodlight SDN controller was used in our implementation.

Communication between the control layer and the infrastructure layer is performed by an OpenFlow protocol [28], which represents SDN standard in the scientific area and specifies mechanisms for its extension. In order to improve a flow processing on WTP, we have extended the OpenFlow protocol. The extension simplifies wireless management in the SDN architecture and unifies wireless and wired management to one control channel. Furthermore, the infrastructure layer is divided into two sublayers for wired and wireless parts. The transport sublayer represents a wired part of the network and radio sublayer for a wireless part of the 802.11 network infrastructure. Transport and radio sublayers are linked via the WTP component. This component represents an edge forwarder, considering a wired part point of view (Transport sublayer). The architecture is primarily focused on 802.11 standard, but the Radio sublayer can be used for other wireless technologies e.g., Bluetooth, 802.15.4. The Radio sublayer’s protocol is untouched which results in ability to use even current devices. Additionally, the behavior of devices in the Radio sublayer is possible to change only via WTP component in accordance with 802.11 standard. 

The architecture components (Figure 1) providing all key functionalities for the management of 802.11 network infrastructure are the following:
Authentication server—provides standard authentication server for WPA2 Enterprise authentication. This component communicates with the SDN Controller via Radius protocol.High-level abstraction framework—provides a high-level abstraction for programmers to simplify the development and improvement of new services to the network. A mapping of high-level functions to the OpenFlow control messages is not introduced in this work. Solutions like [10] already provide some level of abstraction. SDN Controller—represents the main central control component in the architecture due to the fact that the control plane is situated in it. The SDN controller contains applications which provide network services. These applications are on top of the SDN controller and communicate between them. The SDN controller is implementation dependent. The used Floodlight controller allows applications to be built on REST API or as a module. Mobility service—contains decision making for performing handover and modifies the client’s traffic flow to a new WTP. The VAP management is performed by this service. This management contains all VAPs in the network with their position (WTP identification) and statistics.Authentication service—ensures an authenticator role for WPA2 PSK or WPA2 Enterprise authentication. It has to store all encryption keys for all client stations. These keys are distributed to the Encryption and Decryption component (EnDeC). Within this application we have a key management to ensure the mentioned functionality. SDN forwarder—represents a standard SDN OpenFlow forwarder without any modification. It contains a forwarding plan of the network and supports only a wired network. In our case it is Ethernet.Encryption and Decryption component (EnDeC)—is a new component in the architecture for WPA2 encryption and decryption functionality. This functionality was moved from WTP to this new specialized component. This movement is done with the goal to increase security and unload access point. Wireless Termination Point—represents the access point for client stations. This component has restricted 802.11 functionality because some of the 802.11 functions are moved to other components (to SDN controller and EnDeC). The encryption keys are not stored in WTP which results in improved security because despite an attacker having physical access to WTP he is not able to get the encryption keys. The WTP uses a Split-MAC mode [29]. A list with 802.11 functionalities is depicted in Table 1. 

The Authentication server and SDN forwarder components provide standard functionality without modification. The Split-MAC mode divides 802.11 functionalities into the following three components: WTP, SDN controller, EnDeC (Table 1). The WTP builds 802.11 data frames and generates 802.11 control and management frames (e.g., ACK, beacon). This functionality is assigned to WTP in order to unload the SDN controller and forward payload data fast to the client station. To achieve this, it was necessary to extend the OpenFlow protocol in order to set communication between WTP and SDN controller. The communication includes necessary data for the 802.11 network infrastructure management. The management and authentication for network infrastructure are performed by the SDN controller. 

The encryption and decryption of client station payload are performed by EnDeC. EnDeC processes Additional Authentication Data (AAD) headers. AAD provides input parameters for WPA2 setup and protects the integrity of 802.11 frames. A detailed process flow is described in Section 3.3.

### 3.2. OpenFlow Extensions

In this part, the proposed OpenFlow extensions are described. All extensions follow the OpenFlow specification and the implementation details of the extensions are available on Github [30]. The OpenFlow protocol links the forwarding plane with the control plane placed in the SDN controller. The OpenFlow forwarder contains one or more flow tables for packet forwarding. Flow entry consists of match field, counters and a set of instructions to apply to matching packets. The instructions contain actions for packet operations. The communication between the SDN controller and the OpenFlow forwarder is performed via a control channel. In the architecture, we bring 802.11 extensions for these features:
Packet type—provides additional information related to a received packet on a port. It is necessary for correct packet handling within OpenFlow switch packet processing. In the architecture, two types of the packet can be received in the WTP component: 802.11 frames; Ethernet frames. Therefore, it is necessary to distinguish between them. As a result, a new packet type was defined for recognizing Ethernet frames from 802.11 frames.Instructions—contains information for processing the received packet. It contains actions to discard, modify, queue, or forward the packets. There is missing trigger functionality which allows one to generate control messages from WTP to the SDN controller based on received 802.11 frame. For this purpose, we introduce new trigger instruction: generating OpenFlow message based on received 802.11 frames. The generating is important for managing states of the client station and fast handover process. Matches—defines fields of a packet which can be compared within flow entries. New matches are designed for 802.11 frames because original matches were proposed for Ethernet frames. The matches are required for correct evaluation rules against processed packet in the OpenFlow forwarder.Messages—serve to exchange information between the SDN controller and the SDN forwarder. For managing 802.11 functionalities of our architecture we propose new OpenFlow messages to control the wireless part of the network infrastructure. The new OpenFlow messages are developed for communication between the SDN controller, WTP and EnDeC. The OpenFlow messages between the SDN controller and WTP serve for connection management. The OpenFlow messages between the SDN controller and EnDeC distribute the encryption keys for accesses of the client stations into 802.11 network infrastructure. The most important messages are add VAP, remove VAP, probe information, messages for statistics, add key, remove key. There are other messages which help manage the wireless part of the network. 

All these features are focused on bringing efficiency of wireless network management via one control channel and the same packet processing as in the wired part of the network. The other solutions use OpenFlow protocol only for wired part of the network. 

### 3.3. OpenFlow Protocol Process Flow for Wireless Network Part

This subsection describes a process flow for our extension of OpenFlow protocol. It considers processes for message flows between:SDN controller ↔ WTP,SDN controller ↔ EnDeC.

#### 3.3.1. Association Process

Well-functioning association process is a cornerstone for dynamic provisioning of network resources. The association process is depicted in the flow chart (Figure 2). Please note that applications with network services are already considered in the SDN controller for clearer understanding. The association process can be described in the following steps:
(1)As an initial step, a client station sends a *Probe request* frame for access to the 802.11 network infrastructure. The WTP receives it and resends this request to the SDN controller in a *Probe information* message. This message contains a station MAC address. The SDN controller confirms connection permission for the client station request. This permission is given according to compliance with the stations’ MAC address list. In case of denied access to the network, the SDN controller does not generate a message for rejected permission to WTP. If access to the 802.11 network infrastructure is granted, the SDN controller generates unique BSSID for the client station and sends VAP to the WTP. This VAP does not contain a client station IP address because the IP address assignment is performed later. The WTP receives VAP and generates a *Probe response* frame. This *Probe response* frame is sent from the WTP to the client station. The *Probe response* frame BBSID field is set based on VAP that is assigned to the client station. (2)In the next step, the client station sends an *Authentication* frame. The authentication frame content is set to open. The WTP performs the authentication and generates a response for the client station. WPA2 Enterprise authentication is performed later. (3)In the third step the client station is associated with VAP. This association is performed by sending an *Association request* frame to the WTP. The WTP generates an *Association response* frame for the client station. The WTP sends this response to the client station and in parallel, it sends information about results of this step to the SDN controller. This information is sent in an *Association information* message to the SDN controller and provides information related to the wireless part of the network. If the WTP rejects *association request* frame from the client station, the SDN controller immediately deletes the VAP. The functionality of automatic response for the *Authentication* frame and *Association request* frame is situated on WTP because the SDN controller decides about allowed connection in the first step (*Probe request*) and does not perform it again. Additionally, the SDN controller controls the connection through WPA2.(4)In this step, the client station is successfully associated with its VAP on the WTP. The client station needs to authenticate toward an authentication server and is also assigned an IP address. These two processes are independent. The authenticator role is moved to the SDN controller for full control over the authentication and for security improvement. The supplicant and the authentication server roles are the same as in the standard architecture. EAPOL and DHCP communications are forwarded to the SDN controller. This is needed for full control over the network. In the EAPOL case, it is needed because the SDN controller is an authenticator and encryption functionality is moved to the EnDeC component. The SDN controller distributes encryption keys to EnDeC. The traffic forwarding rules are set to WTP after successful client station authentication by the SDN controller. In the DHCP case, all DHCP messages are transferred to the DHCP server via the SDN controller. The DHCP server can be situated within the SDN controller or can be standalone. The SDN controller extracts the client station’s IP address and adds it to VAP management. The same action is done by the WTP.

#### 3.3.2. Handover Process

The handover is performed by the SDN controller that manages the migration of VAP between WTPs. The SDN controller performs handover according to statistics from the WTPs. The handover process is depicted in Figure 3. Our implementation considers signal strength as a key performance parameter for handover but in the future work we will present more sophisticated handover algorithms based on other data like AP load and number of associated client stations. 

The handover process flow is initiated at the SDN controller. The SDN controller generates a *Remove VAP* message to the old WTP (WTP1 in Figure 3) and sends an *Add VAP* to the new WTP (WTP2 in Figure 3). Then the SDN controller changes client station data flows in the wired part of the network to the new WTP. The client station still sends data to its BSSID. The VAP with the same BSSID is moved to the new WTP. The VAP migration is transparent to the client station.

#### 3.3.3. Disassociation Process

This process has various possibilities how can be performed because a *Disassociation* frame is the only announcement and it is not a request type by the IEEE 802.11 standard. This message can be sent by the client station or the WTP. In addition, the client station can disconnect from the network without announcements due to the signal loss. We recognize three scenarios for station disconnection. They are:
(1)A client station disconnects without any announcement. In this case, the WTP must have a mechanism to detect a disconnection of the client station e.g., timer. When the WTP detects client station disconnection, it sends a *Disassociation information* message to the SDN controller. The SDN controller sends a *Remove VAP* message for VAP deletion. (2)A client station announces its disconnection. This scenario has similar behavior as the first scenario without announcement. The difference is that the WTP does not use a mechanism for the client station disconnection. It recognizes disconnection based on the received *Disassociation* frame on the wireless part of the network. In the first and this scenario, the *Remove VAP* message is used for deletion of VAP on the WTP.(3)The SDN controller announces a client station about its disconnection. In this case, the SDN controller decides about client station disconnection based on its internal policy. The SDN controller sends the *Disassociation* message to the WTP. The WTP knows that it has to delete VAP from itself and it also sends the *Disassociation* frame to the client station. 

#### 3.3.4. Add and Update Encryptions Key in EnDeC

The encryption functionality is performed by EnDeC. On the other hand, EnDeC does not enter the authentication process. EnDeC receives encryption key for each client station from the SDN controller. The keys distribution is performed in two scenarios. The first scenario considers client station associating to the network and making successful WPA2 enterprise authentication. The SDN controller has these encryption keys (the authenticator knows master and transient keys). The SDN controller sends transient keys to the EnDeC in an *Add key* message. These transient keys are used to encrypt and decrypt 802.11 traffic. The second scenario considers invalid transient keys due to time out. In this case the SDN controller generates and sends an *Update key* message. 

#### 3.3.5. Data Flow

The architecture includes the EnDeC component for encryption and decryption of 802.11 client station’s traffic. This functionality is removed from the WTP and is fully performed by the EnDeC. The WTP has to forward all data traffic to the EnDeC component which will decrypt it. The next step is to forward data to the network (Figure 4) or encrypt this data for another 802.11 client station in the wireless part of the network (Figure 5). 

Encrypted data has to be transferred between WTP and EnDeC with AAD (Additional Authentication Data) which is also protected against modification. The length of AAD is from 22B to 30B depending on the presence of an Address 4 field (6B) and QoS control field (2B). In our architecture we do not use Address 4. The AAD creates 22B or 24B overhead in the wired part of the network. The 802.11 header is not compatible with Ethernet and therefore, we need to tunnel this data (AAD + data). The L2 tunnel is enough for this purpose. We use the 802.1ah [31] protocol called “Mac in Mac” for L2 tunneling. This protocol creates additional overhead because header has 22B and Ethernet has 14B. The total overhead for transferred data is 30B or 32B between the WTP and the EnDeC. This overhead allows transfer of only 1468B or 1470B data in Ethernet. The architecture can use ICMP protocol to set MTU to one of the mentioned values. Another solution is to use Jumbo frames between the WTP and the EnDeC. 

## 4. Proof-of-Concept Validation

In this section, correctness and performance of our solution is evaluated under Colored Petri Nets and two real world scenarios. Design of our testbed is initially presented. Furthermore, test requirements are summarized for architecture evaluation. According to the summarized test requirements, two critical real-world scenarios were defined with regard to delay sensitive traffic and congestion. Finally, test measurements were performed and evaluated for both scenarios.

### 4.1. Model of Proof of the Architecture Design

Development of our SDN architecture for IEEE 802.11 wireless infrastructure represents the development and implementation of a complex process. In parallel, the behavior of the whole WLAN infrastructure is quite a complex ecosystem. The proposed architecture can be characterized as a concurrent system due to simultaneously executing software components, applications, operations, processes which rely on communication and resource sharing. Colored Petri Nets (CPNs) [32] are a mathematical instrument for modeling and design validation especially for protocol validation and packet switched networks e.g., [33]. The correctness of the communication protocol design can be confirmed by this tool. Therefore, the initial Proof-of-concept of our SDN architecture was done with CPN.

Prior to the implementation of our architecture, we built a model to prove our design. Our aim was to validate all process flows between all the components of the proposed architecture and demonstrate the scalability of the entire network and radio resources control. The model consists of client stations, two WTPs, the SDN controller, and the EnDeC. CSMA/CA mechanism, DHCP message exchange between a client station and a DHCP server and messages for WPA2 authentication were simplified because they are well-known processes, which have been already evaluated. DHCP and WPA2 messages are replaced by the simple request and reply. Our model uses symbolic names for addresses and protocols. 

The model includes two layers. The first layer consists of all SDN architecture components (Figure 6) and their interconnections. The second layer contains a functional description of SDN components. Our extension of OpenFlow is included. This hierarchy was used to simplify the extension of the model with other components. Additionally, all components can be modified without any necessary changes of connections between components. We have validated our design under following standard use cases in 802.11 networks including full message flows between the SDN controller and WTP, the SDN controller and EnDeC. The validation test case includes: association and disassociation processes, add and update encryptions key, seamless handover. The evaluation of the use cases was performed based on Colored Petri Nets properties liveness, boundedness and reachability. Additionally, we verified states of the model against expected states. Finally, our model is available online on Github [30].

#### Evaluation of the Architecture Model

The implemented model of the architecture used two approaches during test scenarios. The first was without a time extension. The model achieved liveness, boundedness and reachability. Frame duplications and losses were random because transitions in the model were randomly activated without any time dependency. Theoretically, new handover could occur before *Add* or *Remove VAP* message was delivered to the WTP component. The randomness could potentially show a deficiency in the architecture design. During the evaluation, the CPN does not show any deficiency in the architecture design. It validates the correctness of proposed communication protocol and there is no blocking state.

The second approach was with the time extension that allows observing the handover influence of frame duplication or frame loss dependency on time. The important time value for model configuration is a delivery time of *Add VAP* and *Remove VAP* messages. The time difference between these messages influences the frame duplication and frame loss. Relations between time for adding VAP and time for removing VAP are shown in Equations (1) and (2):T_Add VAP_ > T_Remove VAP_ ⟶ frame duplication(1)

T_Remove VAP_ > T_Add VAP_ ⟶ frame loss(2)

The count of impacted frames (duplication or loss) can be calculated by the Equation (3). *T_max_* is a maximum delivery time of the first message type and *T_min_* is a minimum delivery time of the second message type. In the equation the message type is interchangeable and represents the worst scenario which can occur. The transfer time *T_transfer_* represents a time of one frame transmission which contains all times (interframe space, ack transfer, etc.) needed for successful frame transmission. A smaller transmission time has a bigger impact on duplicity or loss:Count = (T_max_ − T_min_)/T_transfer_(3)

Except for the scenarios mentioned above, we proposed a scenario for evaluation of handover effects on frames to show that our solution scales into large network. The VAP of one client station is moved between WTP1 and WTP2. The client stations transfer data to the network, and we observe its behavior in frame duplication and loss. 

The scenario duration was set to 10 seconds and handover was performed 98 times during the scenario for one client station. The T_max_ was set to 5 ms and the T_min_ was set to 0.2 ms. The transmission time T_transfer_ was set to 0.329 ms for 1500-byte payload based on standard 802.11g with Tx rate 54 Mbps. The result of Equation (3) was 14 frames and this value was not exceeded during simulation. The results for T_max_ = 5 ms and T_min_ = 0.2 ms are shown in Table 2 and Figure 7, and impacted frames were in the range from 1.32 % to 1.58 %. For T_max_ = 20 ms and T_min_ = 0.2 ms impacted frames achieved an average of 6.26 %. The model does not consider frame retransmissions which results in less frame loss but increases delay.

### 4.2. Testbed of Real Environment Scenario

Our testbed emulates WLAN infrastructure for our defined test scenarios. The testbed was designed to avoid any network or processing power bottlenecks on the hardware or software side. Regular office hardware and software components were used. 

The SDN controller and DHCP server are running on a desktop computer with 32-bit Ubuntu 14.04 operating system. This desktop computer has i7, 2.7 GHz processor and 8 GB RAM. The newest Floodlight version 1.2 was used as the SDN controller. Wireless network part was represented by two MikroTik RB951G-2HnD with an Atheros chipset running with ath9k_htc driver which were used as WTPs (further called APs because we used real routers). Network SSID was set to Inwifi. The test topology is depicted in Figure 8.

The client part was represented by two computers, active and passive one. The stations used D-Link DWA-127, TP-Link TL-WN722N, and Intel Centrino Wireless-N 1000 wireless cards. A passive station was connected to the network in monitoring mode. An active station was sending and receiving generated data. The wireless test measurements were performed under 802.11g standard with maximal throughput up to 54 Mbps. 

The network element of the wired network part was represented by a hub. We know that in real network topology, there is no hub. Our goal was to see the performance of pure SDN architecture with wireless elements avoiding the influence of other peripherals like a switch, router, etc. We also wanted to evaluate our VAP implementation.

Traffic flows were generated with Distributed Internet Traffic Generator (D-ITG) and for the congestion scenario, we used JPerf (https://sourceforge.net/projects/iperf/files/jperf/) JPerf is a GUI interface to iPerf traffic generator suitable for the end to end throughput measurements. D-ITG generates traffic flows for packet switched networks. D-ITG includes models for real-time communication traffic. IPerf is used for active measurements of the maximum achievable bandwidth on IP networks. It measures throughput between two endpoints.

### 4.3. Proposed Scenarios for Real Environment Testing

Test requirements were designed to show the performance of our architecture. In the next step, test scenarios were defined according to test requirements. For evaluation of the SDN WLAN infrastructure, two crucial aspects are client’s mobility (seamless handover) and the highest achievable throughput between a station and an AP. Therefore, two scenarios were defined to show that our proposed solution works well in terms of latency and throughput, with frequently induced handovers, and that unifying management flows into one control channel does not introduce any delay increase. Furthermore, tests were performed under simulation of real-world conditions with interference from the neighboring APs. The proof done under these condition shows how the processes might work and that is why we did not clean it out of the surrounding influences. Within the proof-of-concept validation the whole environment was credible. The Figure of interferences is on Github [30]. In the proof of concept, we used only existing devices without change of MAC or physical layer, which conform with existing 802.11 standards.

In the first scenario, data flows of interactive multiplayer first-person shooter game were used. We wanted an easily reproducible scenario. According to the previous work [34] we used D-ITG model for Quake3 which is one of the most delay sensitive scenarios. Delay sensitivity is defined through the network performance parameters delay and jitter. This scenario is not sensitive to packet loss up to 40% [35]. In parallel, we enforce multiple seamless handovers between testbed APs to investigate handover influence.

Handovers were set according to the pedestrian model [36] as well as Cisco study on Location-Aware WLAN [37]. In the pedestrian model, walking in a passage has a velocity of around 1.0 ± 0.02 m/s. From the Cisco study, a distance among APs should be between 11 and 22 m in most indoor locations for real-time applications. However, they also mention the overlap of wireless cells, which should be around 20%. According to our tests, the longest distance from AP where we had a Cisco mentioned minimum signal level of −67 dBm suitable for real-time application was 17 m. Applying these numbers mathematically, we have calculated an inter-access point distance to 30 m and an estimated cell size of 17 m. The overlap is 24%. When we consider previous calculations, we set the handover interval to 15 seconds. The handover is accomplished with the forced handoff of VAP between APs. In this scenario we are trying to reflect reality, because the Doppler effect is negligible for the 802.11g standard at used modulation and bandwidth. The handovers were repeated six times to exclude possible outliers and obtain results reflecting sufficient statistics. We have performed 8 measurements with 100 seconds duration. Across all the tests we have achieved the average delay of 8.3 ms, jitter of 1.5 ms and packet loss of 0.54 %. In the Figure 9, results from one of the measurements are depicted. By analyzing the plot and logs we found out that the maximum delay is 98 ms, the average delay is 5.2 ms, jitter is 1.2 ms and 0.30 % of packets were dropped. We can also observe that the handover does not introduce any notable delay and all the delay peaks are, to our knowledge, caused by external interferences. 

When we insert our averaged results to MOS formula for a Quake model [35], it returns 4.18 (the scale is 0 to 5, the higher is better). Note that the model was studied for Quake IV and we used it for Quake 3, but there are no significant differences between them in the network part. 

The second scenario is basically a performance evaluation achieved with JPerf, a widely used tool for network performance measurement. The server side was at the desktop with the SDN controller and the client side was on the active laptop (see Figure 8). We repeated this scenario on three different wireless dongles: D-Link DWA-127, TP-Link TL-WN722N, Intel Centrino Wireless-N 1000.

The UDP buffer size was set to 160 kB, which is a default value at the iPerf server. According to the Cisco study [38], the average highest achievable throughput is 25 Mbps on 802.11g at the application layer, which was our target for the end-to-end measurement. We measured the peak congestion with the highest throughput. We set UDP bandwidth on iPerf to 54 Mbps according to a maximum raw data throughput of 802.11g standard. The results shown in Figure 10 are from three different measurements on three different wireless dongles. The handover was enforced every 15 seconds according to the pedestrian scenario and the test duration was 100 seconds. The Figure 10 shows minimal throughput decrease affected by the handover process. We can also observe significant differences in maximum throughput, which can be measured using the provided dongles. This is because of different chipsets inside the dongles and also different implementations of IEEE 802.11 standards. The D-Link with its average 25.9 Mbps is the best. TP-Link, which ended second has average maximum throughput 19.5 Mbps and the last is Intel with 18.9 Mbps. The interesting observation is that in spite of being the worst, the Intel dongle is the most stable without bigger peaks. 

The figure shows only results from one measurement, but we have done 4 measurements on each of the dongles. The average throughputs are shown in Table 3.

## 5. Materials and Methods

Our source codes are available on GitHub [30]. The repository includes source codes for agent and also master with controller. Furthermore, there are also libraries with our own OpenFlow messages done in Loxigen. The router, which runs the agent needs to be OpenWrt supported with the ability to run Click modular router and Open vSwitch. First step is to create OpenWrt image suitable for the router including Open vSwitch and modified wireless network card driver. The next step is cross-compiling Click modular router for our agent. The steps to compile mentioned source codes are in readme files provided in each directory on GitHub. The controller, Floodlight, and master node can be compiled and run on Linux systems or Mac OS. The last step is to firstly run the controller and the master on a PC, and after that the agent on a router. Additionally, the CPN model for architecture verification is available on GitHub [30].

## 6. Conclusions and Future Work

Our aim was to design, develop and demonstrate efficient and unified network management with improved client’s mobility and security for a WLAN infrastructure. Furthermore, our unified network management introduces easier scalability for SDN architecture using multi-controller environment and radio resources control. Utilizing common SDN control channel there are no obstacles for future implementation of distributed controller approach as described e.g., in [13]. Additionally, the AP can be realized on any generic hardware router, where OpenWRT can be installed so we also provide options to highly scale network devices in wired part of the architecture. Our proposed security component EnDeC offloads processing at the AP. It is a unique component compared to other solutions because no one except One Big AP has written about special functions security component. Initially, we followed the state of the technology solutions in order to identify the most efficient principles for managing WLAN infrastructures and defining key performance indicators (KPI). The crucial KPIs for our design were the network delay and jitter for delay-sensitive use cases (e.g., teleconference or online gaming) and the maximal network throughput.

Other SDN frameworks provide the most efficient management over the 802.11 networks infrastructures, but they suffer from the following drawbacks: Multiple control channels which usually reduce architecture transparency and avoid the usage of standardized protocols (e.g., a custom protocol in Odin running at UDP port 2819 in the implementation).They only support partial security solution or do not face security issues at all.They only target high-level abstraction for programmers and are not focused on efficient usage of network resources.They are missing formal verification to prove the correctness of the communication protocol.

Therefore, we introduced a new extension of the OpenFlow protocol. Proposed extensions of the OpenFlow protocol focus on unifying management flows to one control channel. Our solution follows OpenFlow experimenter specifications [28]. Furthermore, embedded encryption is introduced. It supports end-to-end encryption within the SDN network. We defined relations between components and their functionality. 

In a prior stage, the proof-of-concept validation of SDN functional properties of the proposed network architecture was performed. This was done with Colored Petri Nets (CPNs). The CPN design validation clearly demonstrates the correct functionality of our proposal, it’s scalability into large networks and impact of handover on the frame loss. Formal verification was not present in other solutions. This formal verification refers to correctness of VAP concept and proposed protocol. 

The second proof of the concept was performed by architecture implementation under real test conditions (e.g., including interferences from neighboring networks). For this purpose, testbed and two test scenarios were developed. The first scenario aimed for demonstrating the client’s mobility through the functionality of seamless handover for delay sensitive use case. Presented results clearly show handover testing with 8.3 ms average delay and 1.5 ms average jitter through all measurements that outperform the state of the art solutions. Furthermore, the results show that in the handover process our VAP is fully transparent and minimally affects the SDN network delay and jitter. The maximal network congestion at AP was evaluated in the second scenario including the client’s mobility. The experimental results again show SDN network performance achieving almost the maximum possible network throughput for the 802.11g standard. We offer the only solution on which two independent kinds of evaluation have been done: experimental proof of concept; and formal verification of the protocol, states changes and scalability with CPN. 

As future work, we plan to extend our testbed with recently announced OpenFlow-enabled forwarders and new measurements results will be published soon. We are also going to focus on the higher mobility scenarios because it is widely known that 802.11g standards give good performance for high-speed scenarios.

## Figures and Tables

**Figure 1 sensors-19-01880-f001:**
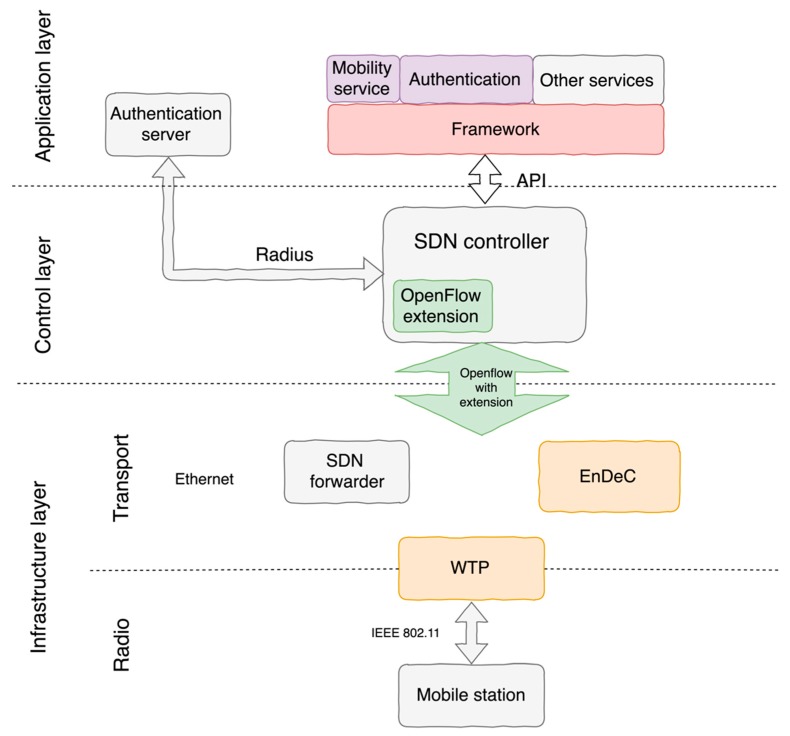
High-level architecture.

**Figure 2 sensors-19-01880-f002:**
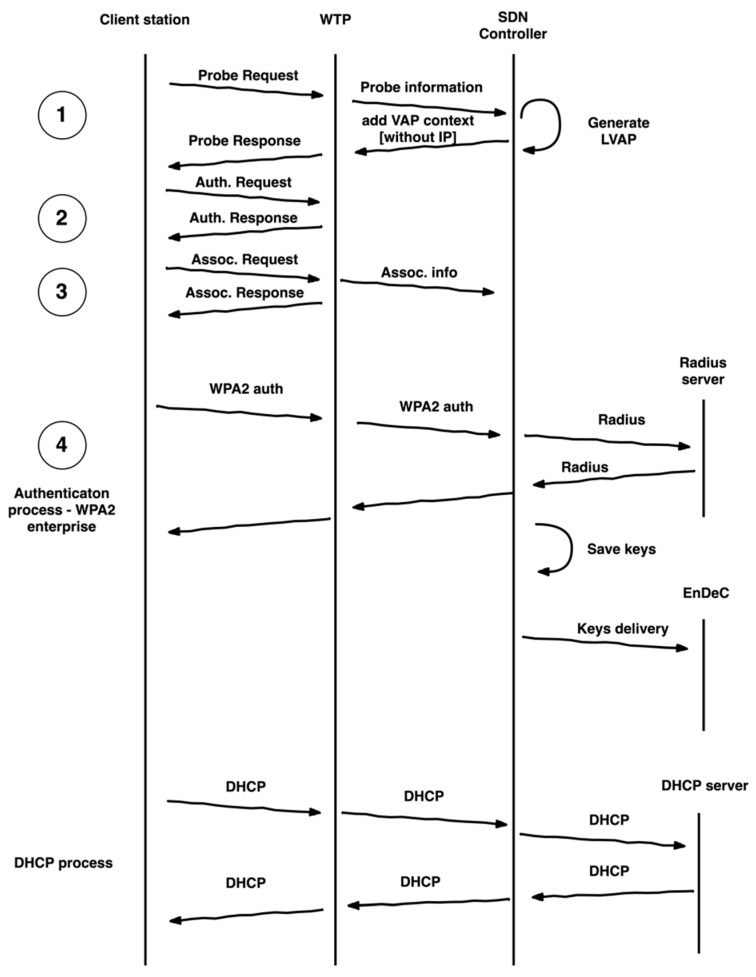
Association process flow.

**Figure 3 sensors-19-01880-f003:**
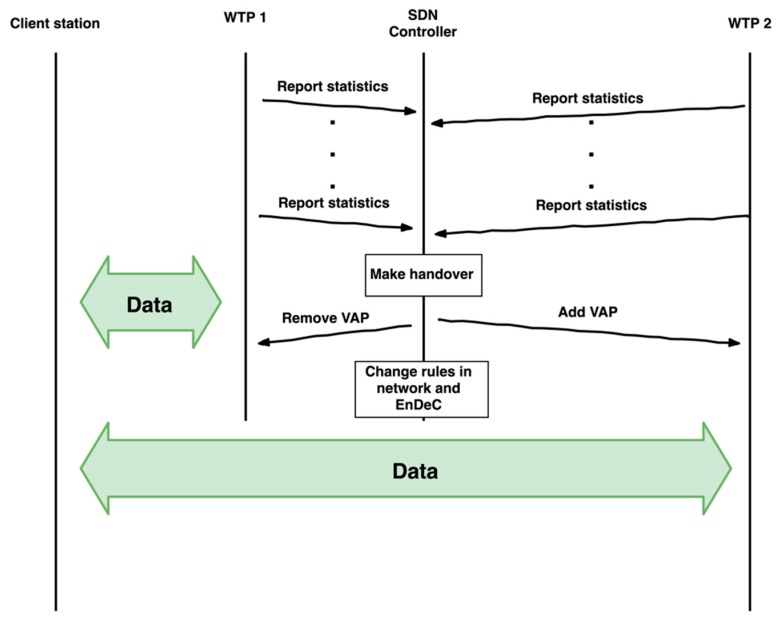
Handover process flow.

**Figure 4 sensors-19-01880-f004:**
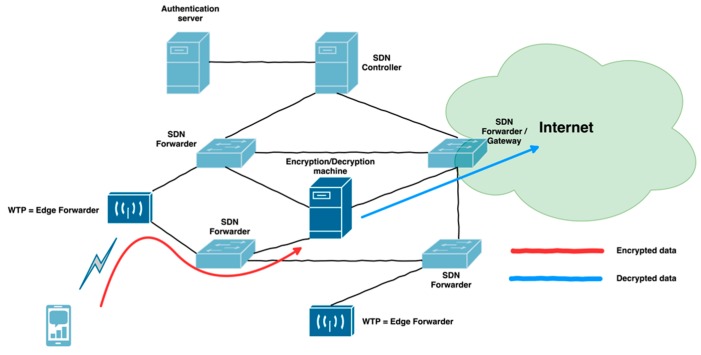
Data flow to the Distribution System.

**Figure 5 sensors-19-01880-f005:**
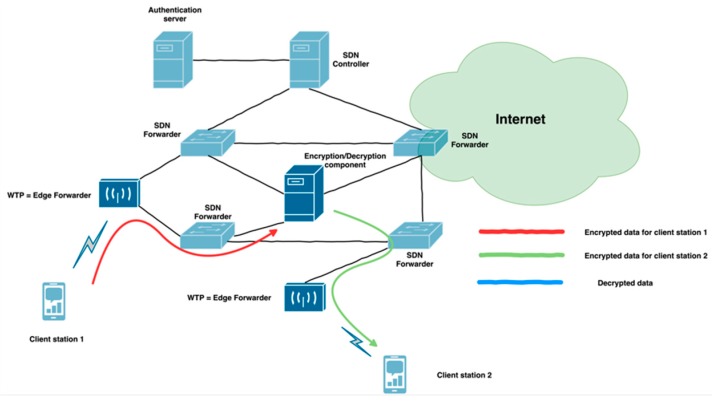
Data flow between two client stations in the same 802.11 network.

**Figure 6 sensors-19-01880-f006:**
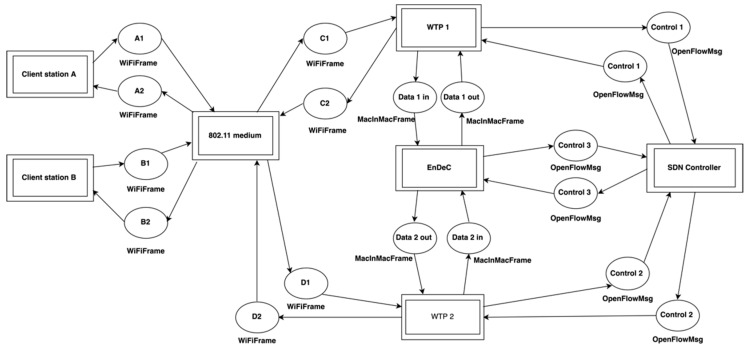
Interconnections of architecture components in Petri nets.

**Figure 7 sensors-19-01880-f007:**
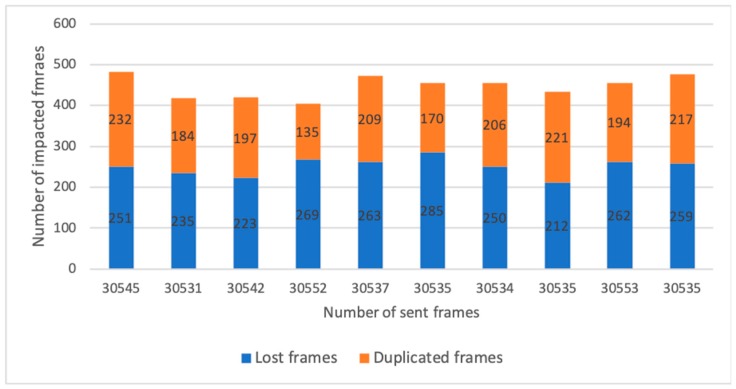
Number of impacted frames during measurements.

**Figure 8 sensors-19-01880-f008:**
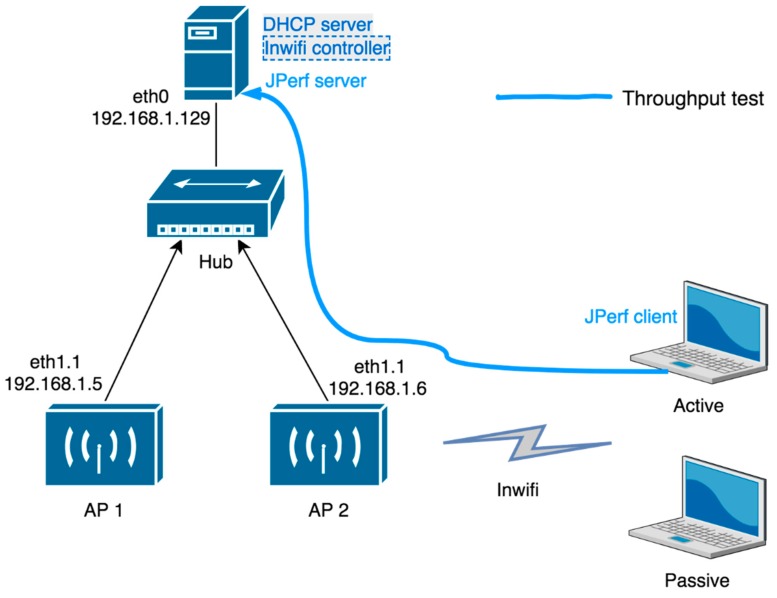
Testbed topology.

**Figure 9 sensors-19-01880-f009:**
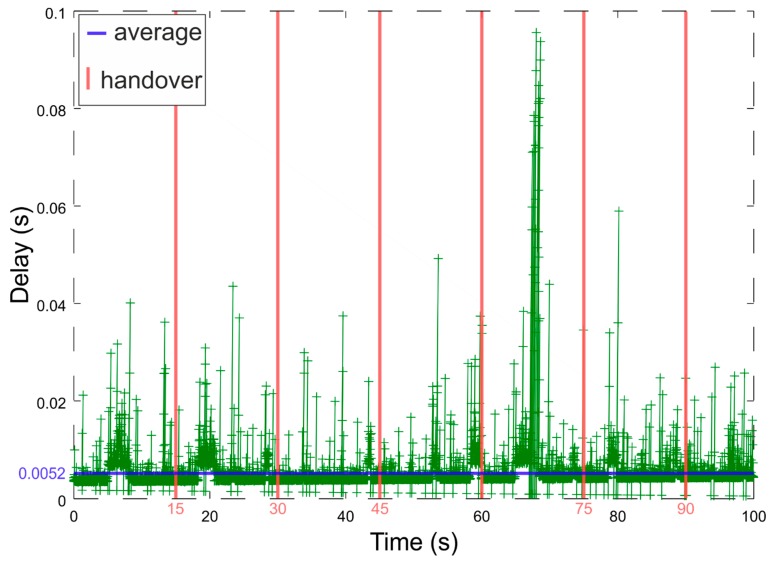
Delay-sensitive scenario measurement.

**Figure 10 sensors-19-01880-f010:**
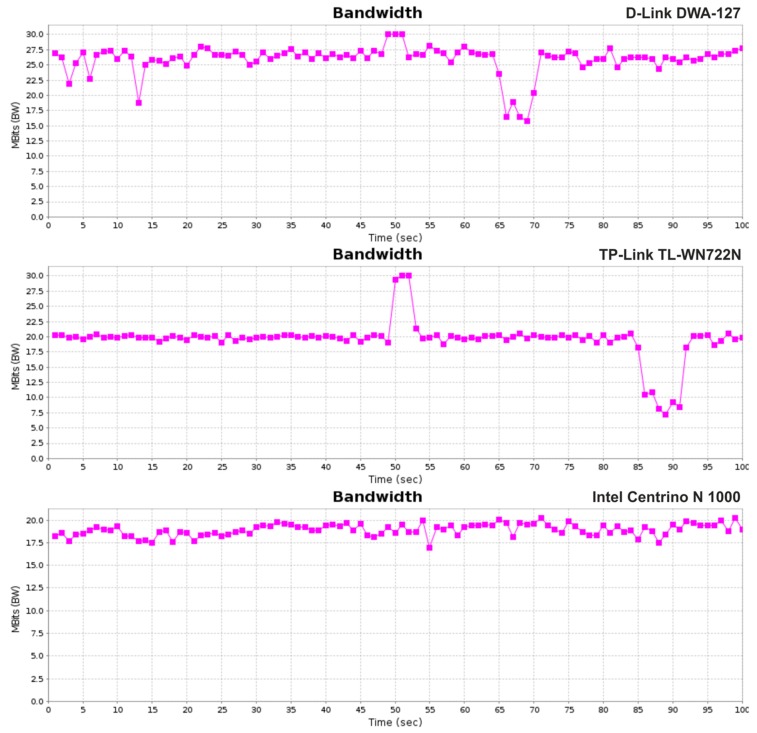
Maximum throughputs for individual measurements.

**Table 1 sensors-19-01880-t001:** Distribution of the 802.11 functionality.

WTP	SDN Controller	EnDeC
Multiple Access Method –CSMA/CA	Authentication	Encryption and decryption of 802.11 frames
Generating 802.11 frames for wireless medium	Control plane (Wireless managing)	Adding 802.11 AAD to client’s data from Distribution System
Building 802.11 frames for wireless medium	Handover decision	

**Table 2 sensors-19-01880-t002:** Measurements of handover impact on transmitted frames

Measurement	Number of Lost Frames	Average Frame Loss during Handover [%]	Number of Handovers with Frame Loss	Number of Duplicity Frames	Average Frame Duplicity during Handover	Number of Handovers with Duplicity Frames	Number of Sent Frames	Impacted Frames during Scenario
1	251	5.12	49	232	4.94	47	30545	1.58
2	235	4.89	49	184	3.2	47	30531	1.37
3	223	5.86	38	197	4.02	49	30542	1.37
4	269	5.38	50	135	3.29	41	30552	1.32
5	263	4.87	54	209	4.75	44	30537	1.54
6	285	5.38	53	170	4.47	38	30535	1.49
7	250	5.68	44	206	4.12	50	30534	1.49
8	212	5.89	36	221	4.51	49	30535	1.41
9	262	5.35	49	194	4.31	45	30553	1.49
10	259	5.63	46	217	5.16	42	30535	1.55

**Table 3 sensors-19-01880-t003:** Average maximum throughputs on three different wireless dongles.

Nr. of Measurement	D-Link	TP-Link	Intel
1	23.2 Mbps	19.5 Mbps	18.6 Mbps
2	25.9 Mbps	18.7 Mbps	18.8 Mbps
3	24.5 Mbps	18.9 Mbps	18.5 Mbps
4	25.1 Mbps	18.5 Mbps	18.9 Mbps
**Average**	24.7 Mbps	18.9 Mbps	18.7 Mbps

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
