# Peer review of "High Performance SDN WLAN Architecture"

_sensors, 2019, doi:10.3390/s19081880_

Reviewer 1 Report

The paper describes a practical solution for anticipating handover using SDN in a WiFi network, also including the authentication phase in the procedure.

The authors certainly spent a substantial amount of work on realising the prototype and carrying out the small-scale demonstrations, but I don't think the paper is ready for publication in a scientific journal, mostly because it provides the reader with little insight on the scientific challenges that have been (or need to be) addressed. Also the performance evaluation is very limited. Overall, the paper looks like a mere report of a basic (though effort- and time-consuming!) engineering exercise.

Some comments:

The related work section can be improved. The authors report a list of solutions mentioning some drawbacks, but it is not clear if and how those are overcome by the proposed solution. For commercial solutions, the authors admit that it is not even possible to know what happens exactly in the black boxes provided by the vendors. I know this process can be frustrating, but if you want to sell a proposed solution as "research" then it is very important to identify exactly which need you are filling. It is not sufficient to say (for instance) Cisco can do handover in 1 second and we did in half a second, so we win. A commercial solution has to deal with a huge amount of matters that do not pop up in a small-scale prototype in a lab: there may well be tons of reasons why your solution once fully-engineered for a production environment does not achieve the same performance measured in a few lab tests.

Regarding the solution itself, the authors invest too much space in implementation details: these are good for a technical report accompanying the product, but not for a scientific paper. An analysis of the technical design choices takes would be much more interesting and useful to the community than an extremely detailed view on the choices themselves, with little justification.

I don't really understand the kind of "validation" that has been done by means of the Petri nets, but this could well be my fault since I am not familiar with this tool.

The testbed measurements are very limited. The most important motivation indicated by the authors for their work is scalability: how can you test this with one AP and two stations.

Author Response

Reply in attachment

Reviewer 2 Report

The paper extended their previous work and the architecture was verified using CPN model and in real environment. The proposed architecture has improved the mobility of mobile clients.

The paper has been overall well written, however the following points should be addressed,

1) It would be useful to summarise the implementation implications. e.g. the new network components required and updates of existing network, compatibilities of existing protocols etc.

2) page 7, the description of mobility service and Authentication service shouldn’t be nested in SDN Controller. They belong to different layers.

3) Some specific corrections:

Page 7, line 295, due to fact… -> due to the fact…

Page 9, line 359, OXM has never been explained.

page 14, line 477, “The client station does not recognize the VAP migration”, should it be “the VAP migration is transparent to the client station”?

Figure 18, there shouldn’t be two eth1.1 in the diagram.

Figure 19, the legend should have covered the green plots.

Author Response

Reply attached

Round  2

Reviewer 1 Report

The overall quality of the paper has certainly improved with respect to the previous submission.

In particular, Section 3 reads much better with the additional text and without the implementation details moved to the github repo.

However, my main concern regarding the contribution being unfocused remains the same.

Let us consider the abstract, for instance.

The authors state that the contributions are:

-  "improvements of the IEEE 802.11 network architecture" -> too generic

- use of SDN -> this is not a contribution per se, but rather a means to achieve a goal

- "communications during devices movements without losing a quality of service" -> this is a clear and nice objective, but scalability is definitely something you want to address

- "the architecture uses concept of a personal access point" -> does this refer to a goal of implementing security over SDN? If this is the case, then 1) it must be re-stated so as to clearly indicate what you want to achieve and 2) it sounds like a self-generated problem: it appears that without SDN this problem does not exist, but since you _decided_ to use it then you have a new problem that must be tackled. If this is the case, then the use of SDN incurs in a trade-off.

Further confusion arises in the testbed performance evaluation. The authors say:

"two scenarios were defined: the first one for the evaluation of minimal network latency; the second one for evaluation of the highest possible network throughput."

How these objectives match the paper contributions?

Both low latency and high throughput are achieved by means of physical/MAC layer technologies, which are well beyond the scope of this paper.

Maybe the authors meant that the scenarios should show that their proposed contribution works well, in terms of latency & throughput, with frequently induced handovers? If this is the case, then the scenarios may be limited anyway since latency/throughput depend on so many other factors.

The testbed results look to me like a proof-of-concept validation of the key contribution of the paper, which is the use of SDN with WiFi stations (for reasons that are not 100% clear to me), but I would not go as far as call these results _performance evaluation_.

Author Response

In attachment

Round  3

Reviewer 1 Report

I am satisfied with the response and review provided by the authors.